# Frozen Stored Teeth: Autogenous Dentin as an Alternative Augmentation Material in Dentistry

**DOI:** 10.3390/bioengineering10040456

**Published:** 2023-04-07

**Authors:** Michael Korsch, Kurt Werner Alt, Frederick Reza Mock

**Affiliations:** 1Dental Academy for Continuing Professional Development, Karlsruhe, Lorenzstrasse 7, 76135 Karlsruhe, Germany; 2Clinic of Operative Dentistry, Periodontology and Preventive Dentistry, University Hospital, Saarland University, Building 73, 66421 Homburg, Germany; 3Private Practice, Center for Implantology and Oral Surgery, Berliner Str. 41, 69120 Heidelberg, Germany; 4Center of Natural and Cultural Human History, Danube Private University, Steiner Landstrasse 124, 3500 Krems-Stein, Austria; 5Private Practice, Practice for Dentistry, Oral Surgery and Implantology, Marienstrasse 9, 66287 Quierschied, Germany; 6Private Practice, 32 Route de Luxembourg, 6130 Junglinster, Luxembourg

**Keywords:** alveolar bone preservation, autogenous, bone graft, dentin, frozen, implant, tooth shell technique, tooth transplantation

## Abstract

Tooth Shell Technique (TST) with the use of autologous dentin has proven to be a suitable method of grafting in the context of lateral ridge augmentation. This present feasibility study aimed to retrospectively evaluate the preservation by lyophilization of processed dentin. Thus, the frozen stored processed dentin matrix (FST: 19 patients with 26 implants) was re-examined with that of processed teeth used immediately after extraction (IUT: 23 patients with 32 implants). Parameters of biological complications, horizontal hard tissue loss, osseointegration, and buccal lamella integrity were used for evaluation. For complications, the observation period was 5 months. Only one graft was lost (IUT group). In the area of minor complications, without the loss of an implant or augmentation, there were two cases of wound dehiscence and one case with inflammation and suppuration (IUT: *n* = 3, FST: *n* = 0). Osseointegration and integrity of the buccal lamella were present in all implants without exception. Statistically, there was no difference between the groups studied for the mean resorption of the crestal width and the buccal lamella. Results of this study show that prepared autologous dentin preserved with a conventional freezer had no disadvantage compared to immediately use autologous dentin in terms of complications and graft resorption in the context of TST.

## 1. Introduction

Implantology represents today a current method for fixed prosthesis. In cases of reduced bone volume, in order to perform a correct implantology, bone augmentation techniques allow to restore the volume. Impressive and steadily increasing is the variety of materials and methods used to compensate for atrophic or inflammatory volume defects [1,2,3,4], depending on the indication. If bone replacement in the form of xenogenic or alloplastic material avoids the harvesting defect, it lacks the osteogenetic and osteoinductive potency of autogenous bone [5,6,7]. This important property of autologous bone is an advantage for the restoration of complex volume defects, which secures it the designation of gold standard [8]. Dentin has similar properties due to its structural similarity to the inorganic matrix and behaves in the same way [9,10,11]. This quality characteristic makes the careless discarding of the biological material of extracted teeth seem increasingly questionable.

Dentin consists of 70% of its wet weight of inorganic hydroxyapatite and amorphous calcium phosphate, and about 20% of organic material; predominantly collagen type I, but also osteocalcin and osteonectin [12], about 8–10% is water [13,14,15,16,17,18]. In bone, the wet weight of the above entities is distributed in percentages of about 45/30/25 [18]. 

So far, autologous bone augmentation has been methodically applied en bloc, ground, or in-shell technique [19]. Promising research results prove comparable success for analogous methods with autologous dentin, which is also applied en bloc, ground, and in Tooth Shell Technique [10,16,20,21]. 

However, what options are available if the tooth extraction and the planned augmentation cannot coincide? Analogous procedure for allogeneic bone freeze-drying [22], e.g., with liquid nitrogen, is too costly for a dental practice. Thus, it seems extremely charming to use a (wisdom) tooth to be extracted for bone augmentation in the same session. However, what to do when extraction and augmentation cannot be performed in the same intervention? A smart solution would be frozen storage of autologous dentin using conventional techniques.

## 2. Materials and Methods

This retrospective study included patients who had undergone lateral maxillary augmentation using the Tooth Shell Technique with immediately processed autogenous dentin and patients who had undergone frozen autogenous dentin in the treatment period between June 2019 and March 2020. Potential patients were identified from the electronic medical record. The study protocol, which was in accordance with the EQUATOR guidelines and the Helsinki Decree, was assessed and released by the Institutional Review Board of the Baden-Württemberg Medical Association (ID: F-2020-068-z). The following conditions applied as inclusion criteria: 

Inclusion criteria: Age of majority applied Tooth Shell Technique for the lateral build-up of the alveolar ridge and at least 4 mm in implant preimplantological region. Furthermore, a non-removable definitive prophetic restoration should be planned and less than four missing teeth. 

Exclusion criteria: Patients under the age of eighteen and patients with periodontal disease, diabetes mellitus (for which an HbA1c value of more than seven percent was assessed), or malignant disease. Furthermore, no pre-irradiated, immunosuppressed patients, or patients on immunosuppressive or antiresorptive medication, as is the case with RANKL inhibitors or medications known as bisphosphonates, were included. The pre-augmentative defects of the alveolar ridge have been less than four millimeters in the area of the planned implant region and/or the definitive prophetic restoration is removable.

Cone beam computed tomography (CBCT) was always performed and evaluated preoperatively. The prerequisite for augmentation was a hard tissue gain of at least 4mm and a circular coverage of the implant of at least 1.5mm bone/autogenous dentin (buccal and oral), so the targeted width of the alveolar ridge should be comparatively at the minimum 6.8 mm with a 3.8 mm diameter implant.

The 42 patients who met the criteria have been differentiated in two groups: **Study group 1:** Freeze-stored teeth (FST): 19 patients (10 female, 9 male) and 22 regions with 26 implants.**Control group 2:** Immediately used teeth (IUT): 23 patients (12 female, 11 male) with 29 regions and 32 implants.

From the medical history we selected these finding variables: Age and sex (Table 1), dental parameters on endodontic and periodontal risks (fractures?), prosthetic conditions, implantological parameters (problems and complications on grafts and implants, inflammatory events, bone level, alveolar crestal width, buccal bone width, and CBCT values over time).

The results of the biological complications, loss of horizontal hard tissue, and condition of the buccal lamella were part of the examination. The complications were assessed as follows: wound dehiscence, inflammation with and without pus formation, significant loss of bone around the implant, and disintegration of the implant. 

Clinical difficulties and complications: Every difficulty and complication suffered by the graft or implant within the observation period was registered. Within this study loss of graft or implant was defined as a severe complication, exposed implant surface, and infection-related massive resorptions. Corresponding to this, the difficulties and complications regarding dehiscence and limited infections not affecting complete osseointegration and complete coverage of the implant surface are defined as non-severe.

Clinical procedure of the graft preparation: The freshly extracted tooth was mechanically cleaned for transfer to an augmentation using a coarse rotating diamond grinding instrument under water cooling (Figure 1a) to separate debris, parodont structures, restorative, and root filling material. A thin dentin shell of the root with a thickness of 1–1.5 mm (Figure 1b) was then extracted using a rotating diamond disk (Frios MicroSaw, Dentsply Sirona Implants, Mannheim, Germany) and simultaneous cooling with sterile saline solution; the remaining tooth material was reduced to granules with particle sizes of 300–1200 µm (Figure 1c,d) using a sterile disposable Chemical cleaning, degreasing, and disinfection of the dentin tray and granulated components were performed by immersion in a solution of sodium hydroxide (0.5 N, 4 mL) and ethanol (20% by volume, 1 mL) (Dentin Cleanser, Kometa Bio, Cresskill, NJ, USA) in a closed sterile tray for 10 min. Sterile gauze was used to absorb our supernatant. Then our graft was poured and rinsed with phosphate-buffered physiological saline (Dulbecco’s Phosphate-Buffered Saline, Kometa Bio, Cresskill, NJ, USA) for three minutes. To release the ost-inducing growth factors and the collagen fiber network, partial dentin demineralization was induced by exposure to 10% EDTA solution (EDTA solution, Kometa Bio, Cresskill, NJ, USA) for three minutes. After this, the material was buffered and rinsed again and immediately used for augmentation or stored frozen after further preparation. The latter is completed after gentle drying at low temperature (below 38 °C) on a hot plate and storage in a sterile gauze at below −18 °C. Thawing for transplantation is also completed at a low temperature (below 38 °C) on a hot plate after moistening it with saline.

### 2.1. General Surgical Procedure of the Tooth Shell Technique (TST)

The perioperative protocol included antibiotics (one day preoperatively and two days postoperatively) with 750 mg amoxicillin three times a day or 300 mg clindamycin in case of penicillin intolerance and 400 mg ibuprofen as analgesic demand medication. Articaine + epinephrine 1:100,000 (Citocartin Sopira^®^, Heraeus Kulzer GmbH, Hanau, Germany) was used for local anesthesia, which all patients received without exception. The incision for the formation of the mucoperiosteal flap was first made crystal on the alveolar ridge or the defect with a vertical relief incision (mesial). Implant bed preparation was performed according to the manufacturer’s protocols for all systems so that each implant could be positioned epicrestally (Figure 2a,b and Figure 3a,b). To cover the defect using the Tooth Shell Technique, the prepared dentin shell is adjusted to the receiver areal and fixed with an osteosynthesis screw (microscrews^®^, Stoma, Emmingen-Liptingen, Germany). The cavities between the shell and the implant were filled with particulated dentin (Figure 2c,d and Figure 3c,d) so that the implant surface was totally covered by bone or autologous dentin (Figure 2c,d and Figure 3c,d), the shell was stably fixed (test with forceps) and the mucoperiosteal flap closed the surgical field without tension after fixation with monofilament, absorbable suture material (Supramid^®^ 5-0, Serag-Wiessner, Naila, Germany). In principle, no bone substitute material or membrane was used in the Tooth Shell Technique.

All implants were exposed and assessed after 3 months. Figure 4c,d show the case in Figure 2 after reopening. Probe measurement points were recorded mesially, distally, orally, and buccally with a periodontal probe. Stability was objectively determined by resonance frequency (Ostell Idx, W&H, Buermoos, Austria). A prosthetic restoration was only made above an implant stability quotient (ISQ) of 60. After a further 2 months, the denture was integrated. The observation period was in total 5 months.

### 2.2. Radiographic Evaluation

A radiological assessment of augmentation and implant placement was performed using a CBCT image (50 × 50 mm FOV, PaX-Duo3D, Orange Dental, Biberach an der Riß, Germany) immediately after implant placement (T1) and for follow-up after 3 months (T2).

The approximal peri-implant bone level, the thickness, and integrity of the buccal lamella and its crestal width were assessed. To assess the approximal peri-implant bone level, the distance from the implant abutment to the implant-bone contact was determined mesiodistally in the T2 examination (highest value each), as was the integrity of the buccal lamella in the Bucco-oral direction (see Figure 4b). The crestal position at T1 was set (Figure 4a). Implant surfaces that did not have a radiopaque structure were the difference between the implant shoulder and the first implant/bone contact (Figure 4b). The buccal lamella and its thickness were determined orthogonally to the implant axis at the level of the implant shoulder (=L0) and 2 mm (=L2) and 4 mm (=L4) below the implant shoulder at T1 and T2 (Figure 4c). The alveolar ridge and its width were determined 2 mm below the implant shoulder in equal planes of T1 and T2 (Figure 4d). Previous studies have evaluated this method of lateral ridge augmentation using CBCT [9,20,23]. The measurements were all performed using Ez3D Plus software (Vatech Co. Ltd., Hwaseong-si, Republic of Korea) by a previously calibrated examiner.

**Figure 4 bioengineering-10-00456-f004:**
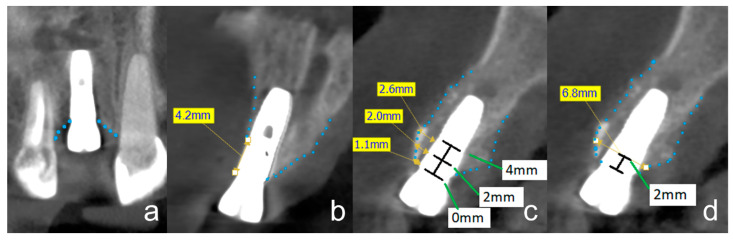
Three-dimensional Cone beam view of implant exposure one quarter after placement. The marked representation circumscribes the jaw area: (**a**) Time of implant exposure; (**b**) Lateral view to assessing the buccal lamella. Implant shoulder to hard tissue (buccal lamella resorption of 4.2); (**c**) Buccal lamella and its width. The lateral tooth shell is unmistakably visible. Measurements were taken at several levels (L0 = 0 mm, L2 = 2 mm, and L4 = 4 mm); (**d**) The dimension of the buccal jaw lamella was measured in the CBCT (L2, width = 6.8 mm).

### 2.3. Osseointegration

Total osseointegration was determined as:-A decrease in peri-implant structures of less than one millimeter at four measurement points.-ISQ values above sixty.-Radiologically covered implant in CBCT.-The vestibular lamella loses less than one millimeter in the CBCT.

### 2.4. Statistical Analyses

All data were entered in Excel and analyzed in IBM SPSS Statistics 22 (SPSS Inc., Chicago, IL, USA) under Windows 7. All evaluations were calculated at the patient, region, and implant levels. Mean values and standard deviations at times T1 and T2 were calculated for Bucco-palatal alveolar ridge width (L2) and buccal lamella width (L0, L2, and L4). The difference between T1 and T2 at the different levels L_x_ was calculated to evaluate the resorption of the buccal lamella. The resorption of the alveolar crest width was calculated at L2. The statistical methods included cross-tabulations with Fisher’s exact tests for categorical data. Mean values were compared by two-sample *t*-tests. 

## 3. Results

TST was performed in 42 patients in 51 jaw sections. The average patient age was 61.4 years. Both subgroups were evenly distributed in terms of age and gender. The implant systems used were ASTRA Tech, Nobel Biocare, and Conelog (ASTRA TECH Implant System™ EV, Dentsply Sirona, New York, NJ, USA; Nobel Biocare, Kloten, Switzerland and CONELOG^®^, ALTATEC GmbH, Wimsheim, Germany).

One severe complication entered during the follow-up period of 5 months. The bone graft was lost on one implant; it was removed at a later date. The patient had two single implants (Figure 5 and Table 2).

### 3.1. Non-Severe Clinical Complications

Apart from the severe complication, three other minor complications occurred in the form of one purulent (FST: *n* = 0; IUT: *n* = 1) and two wound dehiscences (FST: *n* = 0; IUT: *n* = 2); although the purulent complication again affected the patient who also suffered the severe complication.

### 3.2. Radiographic Evaluation

Except for the loss implant, neither vertical loss buccally nor at the mesial and distal shoulders of the implants could be detected on the CBCT images (T1 and T2) (Figure 4a–d). Looking at the course of the alveolar ridge width (Table 3) at the patient level for both groups, the resorption rate was 5% for the FST group and 4% for the IUT group (Table 4 and Table 5). All implants examined were completely covered by hard tissue and the alveolar ridge width did not show any significant difference between the IUT and FST groups in terms of implant, region, or patient level. The same applies to the buccal lamella, which did not show any significant differences when comparing the two groups with regard to the implant, patient, and region level (Table 3, Table 4 and Table 5).

### 3.3. Peri-Implant Tissue Probing, Implant Stability, Osseointegration, and Prosthetic Restoration

Except for the loss augmentation/implant with a severe complication:-All other implants had a probing depth that did not exceed 0.5 mm.-ISQ values between 62 and 87 were registered for all implants and mean values of 74 (IUT group) to 76 (FST group) were determined so that there can be no talk of significant differences.-All implants could be defined as fully osseointegrated as there was no increased probing depth, all ISQ values were above 60 and all implant surfaces were covered with bone and hard tissue.-All implants could be restored with fixed prostheses after implant exposure without any complications.

## 4. Discussion

In a recent study on the Tooth Shell Technique, it was shown that autologous dentin produces comparable results to the Khoury shell technique [20]. A previously known difficulty of autologous dentin is the question of its storability for time-delayed augmentation. Although there are manufacturers of particulator in the Korean-dominated world market who offer commercial storage of autologous dental material in dental banks, this approach does not appear to be universally practicable either from an economic point of view or from the point of view of national law [24]. The present retrospective study aimed to evaluate the feasibility of preserving a prepared dentin matrix: Is frozen storage of autologous dentin possible in dental practice with reasonable effort?

The incidence of clinical complications within the observation period of 5 months was low in both study groups. In group 2 (IUT), the loss of a graft occurred, a severe complication. Furthermore, soft tissue dehiscence was observed at two grafted sites (FST *n* = 0; IUT *n* = 2). Similarly, inflammation was documented at one grafted site during the early healing phase. The incidence of complications is comparable to that of autogenous bone grafts. No significant differences were found when considering the mean resorption of the bucco-oral dimension of the alveolar ridge width between grafting (T1) and re-entry after 3 months (T2). This was 0.46mm in the FST group and 0.42 in the IUT group.

Comparable to the findings of other studies, the majority of resorptions in the present study also take place on the buccal side (L2). These are 0.33 mm (of 0.46 mm (total resorption at implant level)) for the FST group and 0.30 mm (of 0.42 mm (total resorption at implant level)) for the IUT group in the augmentation area. Regarding the incidences of clinical complications and peri-implant bone loss, the results correspond not only to those of previous studies, but also to those of the bone shell technique [23]. The sock-shield technique also confirms good results with autologous dentin within the circumscribed limitation to a single-stage procedure and the same region. The focus here is on preserving the buccal lamella, which is at risk of resorption [25,26].

The degree of osseointegration was determined and evaluated using resonance frequency analysis, and clinical and radiographic assessment of the implant-covering hard tissue. The degree of osseointegration can be determined objectively and reproducibly with a validated method. A resonance frequency analyzer from W&H is used for this purpose. This determines a so-called implant stability quotient (ISQ) as a surrogate parameter for osseointegration [27,28]. According to the manufacturer, ISQ values of more than 60 provide sufficient implant stability for loading by dentures. The study showed that both groups achieved high ISQ values (FST. 76; IUT: 74) so more than adequate osseointegration must be assumed.

The question of whether the process of conventional freezing affects the structure of the dentin matrix has not yet been adequately clarified. Do the osteoinductive growth factors, such as BMPs, IGF-II, and TGF-ß, suffer irreversible damage to their cell [29] and protein structures [30] or even denaturation due to the occasional formation of ice crystals? It is certain that processes with higher freezing rates, e.g. with liquid nitrogen, protect the proteins from crystallization [31]. The most important methods are used in the context of Oocyte cryopreservation freezing [32]. With conventional freezing, a procedure was tested here that ensures the highest possible benefit with justifiable effort in dental practice.

On the other hand, freezing leads to the slowing down of microbial growth [33] and protein-degrading enzymes [34]. Conversely, this theoretically possible impairment of the quality of the frozen product due to the low freezing speed of conventional freezing technology should be reflected in the formation of larger ice crystals from intracellular and extracellular fluid [35,36] and be reflected in a higher failure rate. However, there is a low incidence of clinical complications, a low resorption rate, and sufficient osseointegration of the graft. Thus, it must be concluded that frozen storage of autologous dentin in a conventional freezer in dental practice should not be expected to result in a loss of quality. 

It must be mentioned self-critically that histological preparations and a longer follow-up period are missing. However, this is due to the protocol of simultaneous augmentation and implantation in this study. Ethical reasons do not allow further core drilling and the CBCT data do not allow any conclusions to be drawn about the remodeling of the grafted dentin. However, there is histological evidence regarding the behavior of grafted dentin. These showed slow resorption and de novo formation of bone [10,37,38,39].

With the two-stage procedure shown here, a broadening of the spectrum of indications and methods for augmentation in dentistry, especially dental implantology, is available. Further studies with a longer observation period are certainly desirable, but further development of the techniques is also possible. 

The results encourage a new contextual weighing of older working hypotheses, also in light of new studies.

## 5. Conclusions

The present retrospective study was able to establish the equivalence of results for the use of prepared autologous dentin matrix and prepared, conventionally frozen preserved autologous dentin matrix in the context of transplantation by Tooth Shell Technique. This two-stage procedure by practicable preservation of autologous dentin is an extremely useful technique for both patient and practitioner in all cases where a one-step process is impossible and opens up opportunities for further research approaches. Dentin block may serve as an alternative graft to support horizontal alveolar ridge augmentation. Dentin blocks showed less resorptions than autogenous bone blocks. With regard to the short follow-up period of 5 months, the statements must be viewed with caution.

Although there are manufacturers of particulator in the Korean-dominated global market that offer commercial storage of autologous dental material in dental banks, this approach does not appear to be universally practicable either from an economic point of view or from the point of view of national law.

## Figures and Tables

**Figure 1 bioengineering-10-00456-f001:**
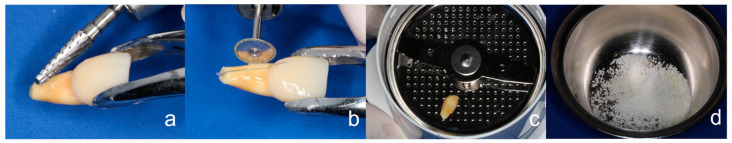
Processing procedure: (**a**) Removal of periodontal structures, root filling, and deposits using rotating diamonds; (**b**) Separation of a dentin shell; (**c**) Particulation of the remaining tooth with the smart grinder; (**d**) Particulated material in a sterile tray.

**Figure 2 bioengineering-10-00456-f002:**
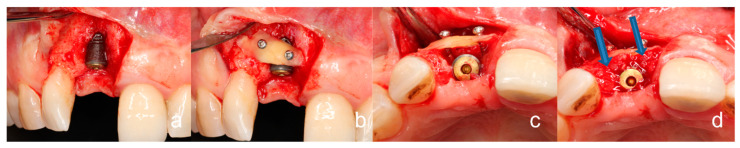
Successive illustration of the Tooth Shell Technique of a tooth that has been stored frozen: (**a**) Lateral view: Implant in region 11. The defect is visible. (**b**,**c**) Multilateral view (lateral and occlusal): The detin shell is stabilized with two osteosynthesis screws on the implant. (**d**) Particulated dentin in the interspace (blue arrows).

**Figure 3 bioengineering-10-00456-f003:**
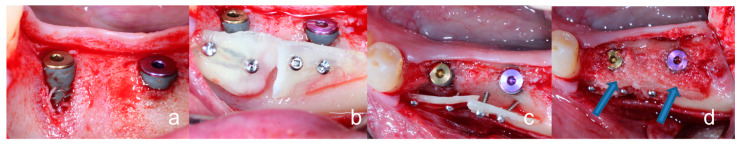
Compilation of the Tooth Shell Technique of a case with an immediately used tooth of another region: (**a**) Lateral view: Implants in region 35 and 36. The lateral defect is visible. (**b**,**c**) Multilateral view (lateral and occlusal): Two detin shells are stabilized with four osteosynthesis screws on the implants. (**d**) Particulated dentin in the interspace (blue arrows).

**Figure 5 bioengineering-10-00456-f005:**
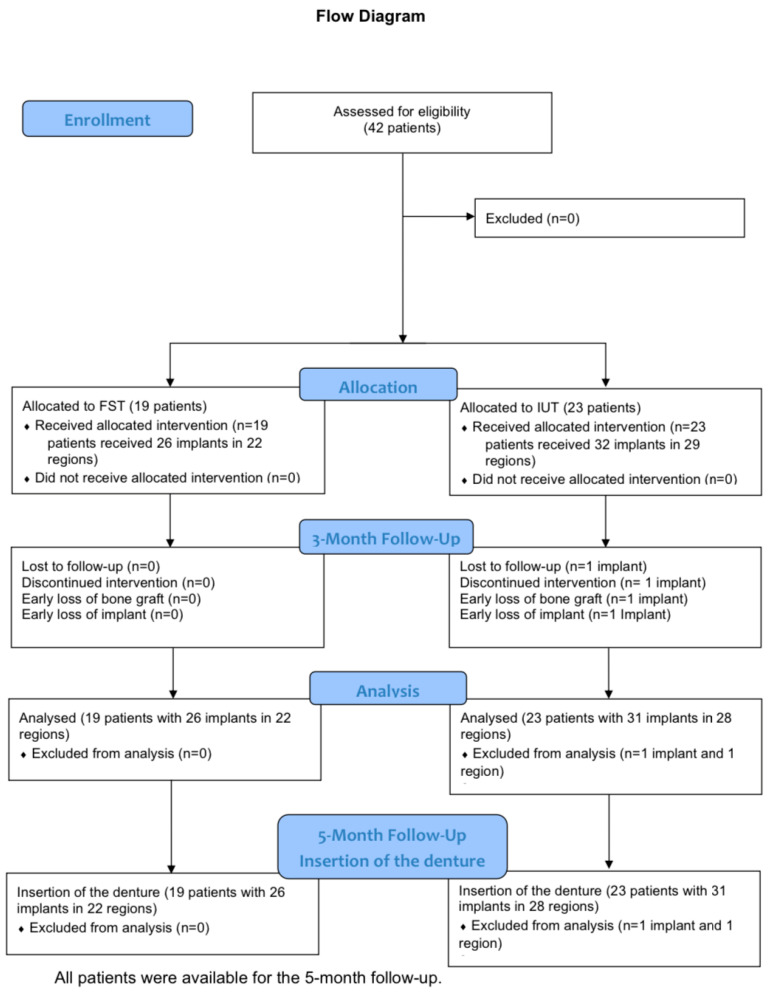
CONSORT flow diagram. The follow-up is reduced in the IUT group in the 3-month follow-up by the implant/augmentation with the severe complication. The patient was further examined due to another implant/augmentation in another jaw region. In the 5-month follow-up, all dentures were inserted.

**Table 1 bioengineering-10-00456-t001:** Starting characteristics of treated patients in augmentation with autogenous dentin.

		Study Groups		Significance
Starting Characteristic	Total	FST	IUT	*p*-Value
**Age (years)**				
**- mean (SD)**	61.4 (10.6)	60.5 (8.2)	62.2 (12.4)	n.s.
**- range**	28–80	38–74	28–80	
**Sex (male)**				
***n* (%)**	20 of 42 (48)	9 of 19 (47)	11 of 23 (48)	n.s.

(SD = standard deviation; FST = frozen stored teeth, IUT = immediately used teeth).

**Table 2 bioengineering-10-00456-t002:** Clinical complications at the patient, region, and implant level.

		Study Groups		Fisher’s Exact Test (2-Sided)
Clinical Complication	Total	FST	IUT	*p*-Value
**Total severe complications**				
***n* (%) on PL**	1 of 42 (2)	0 of 19 (0)	1 of 23 (4)	0.548
***n* (%) on RL**	1 of 51 (2)	0 of 22 (0)	1 of 29 (3)	0.569
***n* (%) on IL**	1 of 58 (2)	0 of 26 (0)	1 of 32 (3)	0.552
**Wound dehiscence**				
***n* (%) on PL**	2 of 42 (5)	0 of 19 (0)	2 of 23 (9)	0.294
***n* (%) on RL**	2 of 51 (4)	0 of 22 (0)	2 of 29 (7)	0.318
***n* (%) on IL**	2 of 58 (3)	0 of 26 (0)	2 of 32 (6)	0.300
**Inflammation (pus)**				
***n* (%) on PL**	1 of 42 (2)	0 of 19 (0)	1 of 23 (4)	0.548.
***n* (%) on RL**	1 of 51 (2)	0 of 22 (0)	1 of 29 (3)	0.569
***n* (%) on IL**	1 of 58 (2)	0 of 26 (0)	1 of 32 (3)	0.552
**Total complications at all**				
***n* (%) on PL**	3 of 42 (7)	0 of 19 (0)	3 of 23 (13)	0.239
***n* (%) on RL**	4 of 51 (6)	0 of 22 (0)	4 of 29 (14)	0.120
***n* (%) on IL**	4 of 58 (5)	0 of 26 (0)	4 of 32 (13)	0.085

(FST = frozen stored teeth, IUT = immediately used teeth, *n* = number, PL = patient level, RL = region level, IL = implant level).

**Table 3 bioengineering-10-00456-t003:** Mean alveolar ridge bone measurements directly after grafting (T1) and at the time of follow-up (T2).

		Study Groups		Two-Sample *t*-Test
Time of Measurement	Mean	FST	IUT	(*p*-Value)
**T1**				
**Mean ucco-oral alveolar ridge width (mm)**				
PL, *n* = 42 (SD)	8.8 (1.6)	8.6 (2.1)	8.9 (1.1)	0.520
RL, *n* = 51 (SD)	8.7 (1.6)	8.5 (2.1)	8.9 (1.1)	0.373
IL, *n* = 58 (SD)	8.7 (1.6)	8.4 (2.1)	8.9 (1.1)	0.212
**Mean buccal lamella width L0 (mm)**				
PL, *n* = 42 (SD)	2.7 (1.0)	2.8 (1.2)	2.6 (0.8)	0.586
RL, *n* = 51 (SD)	2.6 (1.0)	2.7 (1.1)	2.6 (0.9)	0.762
IL, *n* = 58 (SD)	2.6 (1.0)	2.7 (1.1)	2.6 (0.9)	0.733
**Mean buccal lamella width L2 (mm)**				
PL, *n* = 42 (SD)	3.2 (1.0)	3.3 (1.3)	3.1 (0.7)	0.475
RL, *n* = 51 (SD)	3.1 (1.0)	3.2 (1.2)	3.0 (0.7)	0.542
IL, *n* = 58 (SD)	3.1 (1.0)	3.2 (1.2)	3.1 (0.8)	0.581
**Mean buccal lamella width L4 (mm)**				
PL, *n* = 42 (SD)	3.4 (1.3)	3.7 (1.6)	3.2 (1.0)	0.283
RL, *n* = 51 (SD)	3.4 (1.3)	3.6 (1.6)	3.2 (1.0)	0.303
IL, *n* = 58 (SD)	3.4 (1.3)	3.5 (1.5)	3.2 (1.1)	0.359
**T2**				
**Mean bucco-oral alveolar ridge width (mm)**				
PL, *n* = 42 (SD)	8.3 (1.6)	8.1 (2.0)	8.5 (1.1)	0.446
RL, *n* = 50 (SD)	8.3 (1.5)	8.0 (2.0)	8.5 (1.0)	0.244
IL, *n* = 57 (SD)	8.3 (1.6)	7.9 (1.9)	8.5 (1.1)	0.127
**Mean buccal lamella width L0 (mm)**				
PL, *n* = 42 (SD)	2.2 (1.0)	2.2 (1.1)	2.2 (0.9)	0.967
RL, *n* = 50 (SD)	2.2 (1.0)	2.1 (1.1)	2.2 (0.9)	0.888
IL, *n* = 57 (SD)	2.2 (1.0)	2.2 (1.1)	2.2 (0.9)	1.000
**Mean buccal lamella width L2 (mm)**				
PL, *n* = 42 (SD)	2.9 (1.0)	3.0 (1.3)	2.8 (0.8)	0.573
RL, *n* = 50 (SD)	2.8 (1.0)	2.9 (1.2)	2.7 (0.8)	0.576
IL, *n* = 57 (SD)	2.8 (1.0)	2.9 (1.2)	2.8 (0.8)	0.647
**Mean buccal lamella width L4 (mm)**				
PL, *n* = 42 (SD)	3.1 (1.4)	3.3 (1.7)	3.0 (1.0)	0.545
RL, *n* = 50 (SD)	3.1 (1.3)	3.2 (1.7)	3.0 (0.9)	0.634
IL, *n* = 57 (SD)	3.1 (1.3)	3.2 (1.6)	3.1 (1.0)	0.766

(*n* = number, SD = standard deviation, FST = frozen stored teeth, IUT = immediately used teeth, PL = patient level, RL = region level, IL = implant level).

**Table 4 bioengineering-10-00456-t004:** Mean resorption of bucco-oral alveolar ridge bone width and buccal lamella bone plate from T1 to T2.

		Study Groups		Two-Sample *t*-Test
Mean Resorption in mm	Mean	FST	IUT	(*p*-Value)
**bucco-oral alveolar ridge**				
PL, *n* = 42 (SD)	0.44 (0.71)	0.46 (0.69)	0.42 (0.74)	0.874
RL, *n* = 50 (SD)	0.46 (0.76)	0.50 (0.82)	0.42 (0.73)	0.688
IL, *n* = 57 (SD)	0.44 (0.74)	0.48 (0.79)	0.41 (0.70)	0.709
**L0**				
PL, *n* = 42 (SD)	0.45 (0.68)	0.53 (0.76)	0.37 (0.61)	0.460
RL, *n* = 50 (SD)	0.47 (0.68)	0.54 (0.79)	0.41 (0.60)	0.510
IL, *n* = 57 (SD)	0.45 (0.68)	0.50 (0.76)	0.40 (0.61)	0.611
**L2**				
PL, *n* = 42 (SD)	0.31 (0.55)	0.33 (0.50)	0.30 (0.61)	0.871
RL, *n* = 50 (SD)	0.31 (0.55)	0.30 (0.51)	0.31 (0.59)	0.955
IL, *n* = 57 (SD)	0.30 (0.54)	0.31 (0.49)	0.30 (0.58)	0.958
**L4**				
PL, *n* = 42 (SD)	0.35 (0.56)	0.42 (0.45)	0.28 (0.64)	0.450
RL, *n* = 50 (SD)	0.32 (0.57)	0.39 (0.44)	0.26 (0.65)	0.451
IL, *n* = 57 (SD)	0.30 (0.55)	0.38 (0.42)	0.24 (0.63)	0.333

(*n* = number, SD = standard deviation, FST = frozen stored teeth, IUT = immediately used teeth, PL = patient level, RL = region level, IL = implant level).

**Table 5 bioengineering-10-00456-t005:** Ratio of residual ridge width of the bucco-oral alveolar ridge and buccal lamella from T1 to T2 (T2/T1).

		Study Groups		Two-Sample *t*-Test
Ratio from T1 to T2	Mean	FST	IUT	(*p*-Value)
**bucco-oral alveolar ridge**				
PL, *n* = 42 (SD)	0.95 (0.08)	0.95 (0.08)	0.96 (0.08)	0.845
RL, *n* = 50 (SD)	0.95 (0.08)	0.95 (0.09)	0.96 (0.08)	0.632
IL, *n* = 57 (SD)	0.95 (0.08)	0.95 (0.09)	0.96 (0.08)	0.688
**L0**				
PL, *n* = 42 (SD)	0.84 (0.27)	0.82 (0.32)	0.86 (0.23)	0.659
RL, *n* = 50 (SD)	0.83 (0.27)	0.82 (0.32)	0.84 (0.23)	0.825
IL, *n* = 57 (SD)	0.84 (0.28)	0.84 (0.34)	0.84 (0.23)	0.990
**L2**				
PL, *n* = 42 (SD)	0.91 (0.18)	0.90 (0.16)	0.91 (0.20)	0.814
RL, *n* = 50 (SD)	0.91 (0.19)	0.91 (0.17)	0.90 (0.20)	0.906
IL, *n* = 57 (SD)	0.91 (0.18)	0.91 (0.16)	0.91 (0.19)	0.996
**L4**				
PL, *n* = 42 (SD)	0.90 (0.19)	0.86 (0.18)	0.93 (0.20)	0.244
RL, *n* = 50 (SD)	0.91 (0.19)	0.87 (0.17)	0.94 (0.21)	0.245
IL, *n* = 57 (SD)	0.91 (0.19)	0.87 (0.16)	0.94 (0.20)	0.160

(*n* = number, SD = standard deviation, FST = frozen stored teeth, IUT = immediately used teeth, PL = patient level, RL = region level, IL = implant level).

## Data Availability

Data are contained within the article.

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
