# Peer review of "Frozen Stored Teeth: Autogenous Dentin as an Alternative Augmentation Material in Dentistry"

_bioengineering, 2023, doi:10.3390/bioengineering10040456_

Round 1

Reviewer 1 Report

1. please can you addthe following statment in your conclusion or discussion section:

Dentin block may serve as an alternative graft to support horizontal alveolar ridge augmentation. Dentin blocks showed less resorption than autogenous bone blocks.

2. Do you think 5 months of follow-up was enough to draw valid conclusion?

3. tooth shell did work as a bony scaffold and later it will be stay or will replaced with cortical bone?

Author Response

  1. please can you addthe following statment in your conclusion or discussion section:

Dentin block may serve as an alternative graft to support horizontal alveolar ridge augmentation. Dentin blocks showed less resorption than autogenous bone blocks.

>We added the sentence in Conclusion

  1. Do you think 5 months of follow-up was enough to draw valid conclusion?

>5 months is not a long observation period. We also mentioned this as a limitation in the discussion. Thus, the statements in the conclusion must be viewed with reservations. We added the following sentence: “With regard to the short follow-up period of 5 months, the statements must be viewed with caution.”

  1. tooth shell did work as a bony scaffold and later it will be stay or will replaced with cortical bone?

> It is not yet clear whether the dentine shell is permanent or will be resorbed. Studies with longer follow-up periods are therefore necessary.

Reviewer 2 Report

Many thanks for the paper submission. This is a nicely written submission that reports the use of Frozen stored teeth as an alternative augmentation material in dentistry. The paper worths publication but some modifications are required.

1) please order alphabetically the keywords and add two more keywords

2) at line 36 the beginning phrase should be modified,as most of implantology occurs without augmentation procedures. Please modify into

".. Implantology represents today a current method for fixed prosthesis: in cases of reduced bone volume, in order to perform a correct implantology, bone augmentation techniques allow to restore the volume.

please cite the following

Sassano P, Gennaro P, Chisci G, Gabriele G, Aboh IV, Mitro V, di Curzio P. Calvarial onlay graft and submental incision in treatment of atrophic edentulous mandibles: an approach to reduce postoperative complications. J Craniofac Surg. 2014;25(2):693-7. doi: 10.1097/SCS.0000000000000611. PMID: 24621726.   Barone A, Aldini NN, Fini M, Giardino R, Calvo Guirado JL, Covani U. Xenograft versus extraction alone for ridge preservation after tooth removal: a clinical and histomorphometric study. J Periodontol. 2008 Aug;79(8):1370-7. doi: 10.1902/jop.2008.070628. PMID: 18672985.   Graziani F, Gennai S, Cei S, Ducci F, Discepoli N, Carmignani A, Tonetti M. Does enamel matrix derivative application provide additional clinical benefits in residual periodontal pockets associated with suprabony defects? A systematic review and meta-analysis of randomized clinical trials. J Clin Periodontol. 2014 Apr;41(4):377-86. doi: 10.1111/jcpe.12218. Epub 2014 Jan 22. PMID: 24329867.   Chisci G, Fredianelli L. Therapeutic Efficacy of Bromelain in Alveolar Ridge Preservation. Antibiotics (Basel). 2022 Nov 3;11(11):1542. doi: 10.3390/antibiotics11111542. PMID: 36358197; PMCID: PMC9687015.   3) at line 63 please modify the phrase This The retrospective study included patients who had undergone lateral maxillary   into   This retrospective study included patients who had undergone lateral maxillary   4) at line 85 please remothe "in general"   5) figure 2: another figure should be added to the present with the re-opening of the implant   6) another figure compilation with the other technique used whould be added to the manuscript

Author Response

1) please order alphabetically the keywords and add two more keywords

 > We sorted the keywords alphabetically and added two more.

2) at line 36 the beginning phrase should be modified,as most of implantology occurs without augmentation procedures. Please modify into

".. Implantology represents today a current method for fixed prosthesis: in cases of reduced bone volume, in order to perform a correct implantology, bone augmentation techniques allow to restore the volume.

>We have adjusted the sentence in line 36.

3) at line 63 please modify the phrase This The retrospective study included patients who had undergone lateral maxillary   into   This retrospective study included patients who had undergone lateral maxillary   

>We made the adjustment

4) at line 85 please remothe "in general"   

>We made the adjustment

5) figure 2: another figure should be added to the present with the re-opening of the implant   

>Figure 3c and d show the case after reopening.
We added the following sentence: „
Figure 3c and d show the case in figures 2 after reopening.”

6) another figure compilation with the other technique used whould be added to the manuscript.

> The technique was the same in both groups. The difference was that once frozen stored teeth and non-frozen stored teeth were used. Therefore we would not add another figure as this would be redundant.

Reviewer 3 Report

Dear Authors

Missing a declaration that the control group most likely corresponds to the cohort in the publication; M Korsch, 2020. also some Figures

Retrospective study. the control group seems to agree with that in the publication: Korsch, M., Tooth shell technique: A proof of concept with the use of autogenous dentin block grafts. Aust Dent J, 2020. Also part of the figures (Fig. 3 and Fig.1). Lyophylizatione (60) and frozen stored teeth are not se same procedure. BMPs mentioned only in "Discussion" missing quantitively measured. 

 Inclusion and exclusion criteria are described in detail 

the flow chart is useful and clear. 

Conclusion: part of his retrospective study  already published by a single author (M Korsch). In the present study now several (3) authors with the same cohort in the control group. Study M Korsch, 2020 imentioned in References  but not declared in M&M.

Author Response

Missing a declaration that the control group most likely corresponds to the cohort in the publication; M Korsch, 2020. also some Figures

> The patients of the M Korsch 2020 study (22 patients, 27 implants) are not the patients of the control group (23 patients, 32 implants) of the present study. In both, the TST was presented with different goals. The methodology is similar, so the figures are similar. However, as you can see below, the figures are not the same.

Fig. 1 M Korsch 2020

Fig. 1 M Korsch 2023

Fig. 3 M Korsch 2020

Fig. 3 M Korsch 2023

Retrospective study. the control group seems to agree with that in the publication: Korsch, M., Tooth shell technique: A proof of concept with the use of autogenous dentin block grafts. Aust Dent J, 2020. Also part of the figures (Fig. 3 and Fig.1).

>This was previously commented on.

 Lyophylizatione (60) and frozen stored teeth are not se same procedure.

>We changed „lyophilization“ to „frozen storage“

BMPs mentioned only in "Discussion" missing quantitively measured. 

 Inclusion and exclusion criteria are described in detail 

the flow chart is useful and clear. 

Conclusion: part of his retrospective study  already published by a single author (M Korsch). In the present study now several (3) authors with the same cohort in the control group. Study M Korsch, 2020 imentioned in References  but not declared in M&M.

Missing a declaration that the control group most likely corresponds to the cohort in the publication; M Korsch, 2020. also some Figures

> The patients of the M Korsch 2020 study (22 patients, 27 implants) are not the patients of the control group (23 patients, 32 implants) of the present study. In both, the TST was presented with different goals. The methodology is similar, so the figures are similar. However, as you can see below, the figures are not the same.

Fig. 1 M Korsch 2020

Fig. 1 M Korsch 2023

Fig. 3 M Korsch 2020

Fig. 3 M Korsch 2023

Retrospective study. the control group seems to agree with that in the publication: Korsch, M., Tooth shell technique: A proof of concept with the use of autogenous dentin block grafts. Aust Dent J, 2020. Also part of the figures (Fig. 3 and Fig.1).

>This was previously commented on.

 Lyophylizatione (60) and frozen stored teeth are not se same procedure.

>We changed „lyophilization“ to „frozen storage“

BMPs mentioned only in "Discussion" missing quantitively measured. 

 Inclusion and exclusion criteria are described in detail 

the flow chart is useful and clear. 

Conclusion: part of his retrospective study  already published by a single author (M Korsch). In the present study now several (3) authors with the same cohort in the control group. Study M Korsch, 2020 imentioned in References  but not declared in M&M.

Round 2

Reviewer 2 Report

Many thanks for the author modification of the paper, however the paper requires some modifications in order to proceed to publication.

1) the introduction requires a more comprehensive review of the main recent publication regarding materials for bone regeneration. The phrase that was suggested was taken from the following articles:

  Barone A, Aldini NN, Fini M, Giardino R, Calvo Guirado JL, Covani U. Xenograft versus extraction alone for ridge preservation after tooth removal: a clinical and histomorphometric study. J Periodontol. 2008 Aug;79(8):1370-7. doi: 10.1902/jop.2008.070628. 

Graziani F, Gennai S, Cei S, Ducci F, Discepoli N, Carmignani A, Tonetti M. Does enamel matrix derivative application provide additional clinical benefits in residual periodontal pockets associated with suprabony defects? A systematic review and meta-analysis of randomized clinical trials. J Clin Periodontol. 2014 Apr;41(4):377-86. doi: 10.1111/jcpe.12218. Epub 2014 Jan 22.

Chisci G, Fredianelli L. Therapeutic Efficacy of Bromelain in Alveolar Ridge Preservation. Antibiotics (Basel). 2022 Nov 3;11(11):1542. doi: 10.3390/antibiotics11111542

As previously requested in the previous revision, please cite these papers  at line 42

2) The authors are quite confused regarding the number of the figures.

the figure 3c and 3d only report figure of cbct scan, but not the re-opening of the case. Instead, the figure 2d reports the re-opening of the case, bu only in the occlusal view: as in the figure 1a the case is presented in the frontal view, the authors should add a figure that represents a frontal view of the re-opening.

3) the authors at the point 6  answer "The technique was the same in both groups. The difference was that once frozen stored teeth and non-frozen stored teeth were used. Therefore we would not add another figure as this would be redundant."

Dear authors, the role of the reviewers is to evaluate if a figure is redundant or not: and in this case you need to present a case with figure compilation with the other technique (both frozen teeth and non-frozen teeth), if you documented the cases as your reported in the manuscript.

Author Response

1) the introduction requires a more comprehensive review of the main recent publication regarding materials for bone regeneration. The phrase that was suggested was taken from the following articles:

  Barone A, Aldini NN, Fini M, Giardino R, Calvo Guirado JL, Covani U. Xenograft versus extraction alone for ridge preservation after tooth removal: a clinical and histomorphometric study. J Periodontol. 2008 Aug;79(8):1370-7. doi: 10.1902/jop.2008.070628. 

Graziani F, Gennai S, Cei S, Ducci F, Discepoli N, Carmignani A, Tonetti M. Does enamel matrix derivative application provide additional clinical benefits in residual periodontal pockets associated with suprabony defects? A systematic review and meta-analysis of randomized clinical trials. J Clin Periodontol. 2014 Apr;41(4):377-86. doi: 10.1111/jcpe.12218. Epub 2014 Jan 22.

Chisci G, Fredianelli L. Therapeutic Efficacy of Bromelain in Alveolar Ridge Preservation. Antibiotics (Basel). 2022 Nov 3;11(11):1542. doi: 10.3390/antibiotics11111542

As previously requested in the previous revision, please cite these papers  at line 42

> We have inserted the requested studies.

2) The authors are quite confused regarding the number of the figures.

the figure 3c and 3d only report figure of cbct scan, but not the re-opening of the case. Instead, the figure 2d reports the re-opening of the case, bu only in the occlusal view: as in the figure 1a the case is presented in the frontal view, the authors should add a figure that represents a frontal view of the re-opening.

> When exposure of the implant (re-opening), we always proceed in a minimally invasive manner. This means that the osseosynthesis screws are removed transmucosally. The lateral dentin lamella is never shown here. Therefore, when re-opening, we unfortunately cannot show a case in which the dentin lamella can be seen from the lateral side. At time of implant exposure, the peri-implant tissue is assessed with a probe.

3) the authors at the point 6  answer "The technique was the same in both groups. The difference was that once frozen stored teeth and non-frozen stored teeth were used. Therefore we would not add another figure as this would be redundant."

Dear authors, the role of the reviewers is to evaluate if a figure is redundant or not: and in this case you need to present a case with figure compilation with the other technique (both frozen teeth and non-frozen teeth), if you documented the cases as your reported in the manuscript.

> We have added a series of images of a case where the tooth was used immediately (non-frozen tooth).